# Early immune response against *Fonsecaea pedrosoi* requires Dectin-2-mediated Th17 activity, whereas Th1 response, aided by Treg cells, is crucial for fungal clearance in later stage of experimental chromoblastomycosis

Isaque Medeiros Siqueira[1], Marcel Wüthrich[2], Mengyi Li[2], Huafeng Wang[2], Lucas de Oliveira Las-Casas[1], Raffael Júnio Araújo de Castro[1], Bruce Klein[2,3,4], Anamelia Lorenzetti Bocca[5]*

**1** Molecular Pathology Post-Graduate Program, School of Medicine, University of Brasília, Brasília, Brazil, **2** Department of Pediatrics, School of Medicine and Public Health, University of Wisconsin-Madison, Wisconsin, United States of America, **3** Department of Internal Medicine, School of Medicine and Public Health, University of Wisconsin-Madison, Wisconsin, United States of America, **4** Department of Medical Microbiology and Immunology, School of Medicine and Public Health, University of Wisconsin-Madison, Wisconsin, United States of America, **5** Department of Cell Biology, Institute of Biological Sciences, University of Brasília, Brasília, Brazil

* albocca@unb.br

## Abstract

Chromoblastomycosis (CBM) is a chronic worldwide subcutaneous mycosis, caused by several dimorphic, pigmented dematiaceous fungi. It is difficult to treat patients with the disease, mainly because of its recalcitrant nature. The correct activation of host immune response is critical to avoid fungal persistence in the tissue and disease chronification. CD4+ T cells are crucial for the development of protective immunity to *F. pedrosoi* infection. Here, we investigated T helper cell response dynamics during experimental CBM. Following footpad injection with *F. pedrosoi* hyphae and conidia, T cells were skewed towards a Th17 and Th1 phenotype. The Th17 population was the main Th cell subset found in the infected area during the early stages of experimental murine CBM, followed by Th1 predominance in the later stages, coinciding with the remission phase of the disease in this experimental model. Depletion of CD25+ cells, which leads to a reduction of Treg cells in the draining lymph node, resulted in decline in fungal burden after 14 days of infection. However, fungal cells were not cleared in the later stages of the disease, prolonging CBM clinical features in those animals. IL-17A and IFN-γ neutralization hindered fungal cell elimination in the course of the disease. Similarly, in dectin-2 KO animals, Th17 contraction in the course of experimental CBM was accompanied by fungal burden decrease in the first 14 days of infection, although it did not affect disease resolution. In this study, we gained insight into T helper subsets' dynamics following footpad injections of *F. pedrosoi* propagules and uncovered their contribution to disease resolution. The Th17 population proved to be important in eliminating fungal cells in the early stages of infection. The Th1 population, in turn, closely assisted by Treg cells, proved to be relevant not only in the elimination of fungal cells at the

**Data Availability Statement:** All relevant data are within the manuscript and its Supporting Information files.

**Funding:** Funding were provided from Brazil by Coordination of Superior Level Staff Improvement – CAPESIMS (Coordenação de Aperfeiçoamento de Pessoal de Nível Superior), and Federal District Research Foundation – FAPDFALB (Fundação de Apoio à Pesquisa do Distrito Federal – FAPDF project no. 193.496/2009) and from USA by R01 AI093553MW, R01 AI035681 BK and R01 AI040996 BK. The funders had no role in study design, data collection and analysis, decision to publish, or preparation of the manuscript.

**Competing interests:** The authors have declared that no competing interests exist.

beginning of infection but also essential for their complete elimination in later stages of the disease in a mouse experimental model of CBM.

## Author summary

Chromoblastomycosis is a chronic subcutaneous infection caused by several dimorphic, pigmented dematiaceous fungi. CD4+ T cells modulations are crucial for the proper immune response against this fungal infection and play a key role in CBM resolution in a self-healing mouse model. In this work we report Th17 cells as being the main CD4+ sub-population in the infected area during the early stages of experimental murine CBM, followed by Th1 predominance in the later stages, coinciding with the remission phase of the disease in this experimental model. Depletion of CD25+ cells resulted in fungal burden reduction after 14 days of infection, but it compromised fungal clearing in later stages of the disease, prolonging CBM clinical features in those animals. *In vivo* analysis with IL-17A and IFN-γ neutralization hindered fungal cell elimination in the course of the disease. Dectin-2 deficiency was associated with impairment of Th17 response and fungal control in the early phase of CBM but did not affect disease resolution. In this study, we gained insight into T helper subsets' dynamics following footpad injections of *F. pedrosoi* fungal cells and uncovered their contribution to disease resolution.

## Introduction

Invasive fungal infections are a growing threat to public health, and global warming, including climatic oscillations, may be causing the selection of new environmental fungal species that have acquired thermotolerance, a key step toward pathogenesis in humans [1]. In immune-compromised individuals, fungi can establish severe disease, which may require treatment for a lifetime. Besides, current diagnostic techniques and therapy options are limited [2,3].

Chromoblastomycosis (CBM) is typically not lethal but often causes a chronic subcutaneous infection. Its complications can lead to the destruction of lymphatic organs, hyperplasia, and eventually, the amputation of affected limbs, increasing morbidity levels among patients [4]. CBM is considered one of the most challenging mycoses to treat, mainly due to its recalcitrant nature, especially in severe clinical conditions. Treatment usually consists of long periods of antifungal therapy, commonly associated with physical therapies such as surgery, cryotherapy, and thermotherapy [5,6].

The importance of cell-mediated immunity in host protection has been described for various fungal infections, such as cryptococcosis [7], paracoccidioidomycosis [8], histoplasmosis [9], blastomycosis [10] and candidiasis [11]. However, the cellular immune response in CBM has been poorly studied, requiring further investigation.

Immunity against pathogens necessarily involves the recognition of pathogen-associated molecular patterns (PAMPs) via pattern recognition receptors (PRRs), triggering a signaling cascade capable of initiating and directing the responses of both innate and adaptive immunity. In fungal infections, such responses are mediated primarily by members of the C-lectin-type receptor family [12]. CLRs are part of a heterogeneous superfamily of transmembrane proteins characterized by a C-type lectin domain [13], remaining in most fungal species capable of causing disease in humans. These receptors recognize the major carbohydrate structures present in the cell wall of fungi, including β-glucans and mannan [14]. Among CLRs capable

of signaling the recognition of fungal structures are dectin-1, dectin-2, mannose receptor (MR), MCL, DC-SIGN, and mincle [15]. While dectin-1 recognizes β-glucans, the other receptors may bind to a variety of mannose-based structures found in the mannan layers of the fungal cell wall.

Critically required for the orchestration of antifungal responses, CD4+ T cell-mediated response improves host defense to *F. pedrosoi* infection [16]. *In situ* studies on CBM patients, as well as in experimental models, show a correlation between Th2 response and disease severity, together with a high fungal load. The milder form of the disease is related to a Th1 profile, with high IFN-γ production, lower levels of IL-10, and lower fungal burden. In a Th1 dominant response, granulomas are more compact and better organized, resulting in mild lesions, usually in the form of plaques [17,18].

In addition to Th1 and Th2 response, T lymphocyte subpopulations such as Treg and Th17 also play relevant roles in establishing the protective immune response to fungi. Tregs are characterized by the high expression of CD25 and Foxp3 transcription factor, acting both in the modulation of the immune response against pathogens and in the control of the immune response to self and non-self antigens [18–20]. Vital elements in immune tolerance, Tregs may be originated in the thymus (natural) or be generated in the periphery after multiple antigenic stimuli or in so-called tolerogenic (induced) conditions [21]. These cells exert their function by releasing inhibitory cytokines such as IL-10 and TGF-β [20]. High concentrations of TGF-β in the presence of mediators such as retinoic acid drive the immune response to the development of regulatory T cells.

On the other hand, low concentrations of TGF-β associated with pro-inflammatory cytokines such as IL-1β, IL-6 and IL-21 and IL-23 allow the differentiation of CD4 + T cells into Th17, inducing RORγT transcription factor [22–24]. Th1, Th2, Th17, and Treg orchestration in the course of the fungal disease can be a crucial element in the control and resolution of those infections. An unbalanced response may favor the persistence of pathogens in infected tissues, worsening the patient's clinical condition.

In this study, we show T lymphocytes with regulatory profile polarization in the popliteal lymph node of a self-healing animal model of CBM, while Th17 was predominant in the infected area during the early stages of experimental murine CBM. The Th17 profile was then followed by Th1 popularization in later stages, coinciding with the remission phase of the disease in this experimental model. *In vivo* depletion of CD25+ cells leads to Treg cells' impairment in draining lymph nodes (dLN), reflected by a reduction in fungal burden after 14 days of infection. Furthermore, fungal cells were not cleared in later stages of the disease, prolonging CBM clinical features in those animals. *In vivo* analysis with IL-17A and IFN-γ neutralization hindered fungal cell elimination in the course of the disease. Similarly, in dectin-2 KO animals, Th17 contraction in the course of experimental CBM showed fungal burden impairment in the first 14 days of infection, although it did not affect disease resolution.

## Materials and methods

### Fungal strain and infection

*F. pedrosoi* (ATCC 46428) was maintained in Sabouraud Dextrose Agar medium (SDA, Himedia) supplemented with 100 mg/ml of Chloramphenicol at 37˚C, as described previously [25]. Purified conidia and hyphae were obtained by growing virulent *F. pedrosoi* propagules in potato dextrose (PD) medium in a rotary shaker (180 rpm) at 37˚C for 7–14 days. At that time, culture suspensions containing conidia and hyphal fragments were first filtered in sterile fiberglass to remove large hyphae clumps. The filtrate was subjected to successive filtrations through 70 μm and 40 μm cell strainers (BD). Retained hyphae in the 40 μm cell strainer

(measuring 40 to 70 μm) were re-suspended in phosphate-buffered saline (PBS) and centrifuged twice at 1000 g, providing more than 98% of purified hyphae. The filtrate containing conidia and small hyphal fragments from the 40 μm cell strainer was further filtered using a 14 μm filter paper (J. Prolab, Brazil), and centrifuged twice at 3000 g, yielding a cell suspension containing at least 98% purified conidia. Fungal propagules were obtained by mixing purified hyphae and conidia at a 1:3 rate. Muriform cells were obtained using a chemically defined medium (CDM) described by Mendoza *et. al.* (1993) [26], with the following composition: glucose (30g/l); $NaNO_3$ (3g/l); $K_2HPO_4$ (1g/l); $MgSO_4 \cdot 7H_2O$ (0.5g/l); $FeSO_4 \cdot 7H_2O$ (0.01g/l); $NH_4Cl$ (0.265g/l); thiamine, (0.003g/l); and $CaCl_2$ (0.011g/l), pH 2.5. Briefly, mycelium fragments from PD were inoculated into an unacidified version (pH 6.5) of the CDM supplemented with 0.1% (w/v) yeast extract. After 7 days at 37°C and 180rpm, an aliquot of this culture (1%, v/v) was inoculated into the CDM for another 7–14 days. The suspension containing *F. pedrosoi* muriform cells was then filtered through a 40 μm cell strainer, yielding more than 90% of purified muriform cells. Live and purified fungal cells were finally counted in a hemocytometer using trypan blue dye and then inoculated into experimental animal footpad at $2x10^7$ live fungal cells per mL (50 μl per foot).

### Animals and experimental design

C57BL/6 male mice, aged 6 to 8 weeks, were purchased from Jackson Laboratories (Bar Harbor, ME, USA) and maintained under appropriate conditions with water and ad libitum feed in the Microbial Science Building, University of Wisconsin—Madison, USA. Dectin-2 KO male mice, 6 to 8 weeks old, were obtained through breeding in the University of Wisconsin's laboratory. IFN-γ KO male mice, 6 to 8 weeks old, were kindly provided by Dr. Milton Adriano Pelli de Oliveira and kept in University of Brasília facilities.

For the establishment of the murine CBM, 4 to 5 animals per experimental group were infected in the plantar cushion with fungal propagules containing fragments of hyphae and conidia of *F. pedrosoi*. Right hind footpad was inoculated with 50 μl of the fungal solution containing $1x10^6$ fungal cells.

Every three days the infected area was measured with the aid of a caliper and at 7, 14, 21 and 28 days after infection, animals were euthanized by prolonged exposure to $CO_2$, followed by surgical excision of plantar cushion and popliteal lymph node for future analyses aiming at the identification of T helper lymphocytes in the course of CBM. Quantification of colony-forming units (CFUs) was performed by plating homogenized infected tissue from footpad.

### Ethics statement

All procedures involving animals carried out in the USA were approved by the Animal Care Committee of the University of Wisconsin—Madison, following the guidelines for Care and Use of Laboratory Animals issued by the National Institutes of Health in USA (Protocol n°. M00969). Procedures carried out in Brazil were approved by the Ethics Committee for Scientific Studies of the University of Brasilia following the Brazilian Council for the Control of Animal Experiments (CONCEA) guidelines on the use and care of laboratory animals (UnBDoc n°. 135976/2014).

### Histopathology and fungal burden

Small fragments of infected tissue were fixed, dehydrated and embedded in paraffin to evaluate lesion progression. Serial sections were made and stained with hematoxylin and eosin (HE). Infected tissue from each experimental animal was also homogenized in PBS (pH 7.2) and then plated onto SDA medium, supplemented with 100 mg.l$^{-1}$ Chloramphenicol and cultivated

at 37˚C for seven days. Fungal burden was then measured by CFUs of *F. pedrosoi*. Results were expressed as the number of CFUs ± standard error of the mean (SEM).

## Lymphocyte isolation from draining lymph node (dLN) and footpad

The popliteal lymph node was collected and homogenized in a 40μm cell strainer with the aid of a syringe plunger, washed with 5 ml of simple RPMI and centrifuged at 300g for 5 min. Cells were then resuspended in 1mL of RPMI supplemented with 10% FBS and filtered again in a 40μm cell strainer.

Tissue removed from the footpad of mice was cut into small pieces and submitted to 5ml of collagenase D solution (Roche) at 1mg per ml of collagenase buffer containing: HEPES (2.39 g/l), KCl (0.37 g/l), $MgCl_2$ (0.20 g/l), $CaCl_2$ (0.20 g/l) and NaCl (14.09 g/l). After 1 hour of incubation at 37˚C in petri dishes, tissue fragments were homogenized in a 70 μm cell strainer and washed with 5 ml of regular RPMI. Collagenase activity was stopped by the addition of 1ml EDTA at 50mM and centrifuged at 300g for 5 minutes. Pellet was resuspended in 5 ml of simple RPMI, filtered in a 70μm cell strainer and then centrifuged once more. Finally, cells were resuspended in 1 ml of supplemented RPMI. Lymphocytes were then quantified in a Neubauer chamber, and cell viability was measured with trypan blue. Samples were processed individually, per animal.

## Surface and intracellular staining

For labeling, only surface molecules, 200μl aliquot of the cell suspension described above was packed into a 96-well U-bottom culture dish. The plate was centrifuged at 300g and, after discarding the supernatant, a cocktail was added, with each well containing: 100μl Brilliant Stain Buffer (BD biosciences), 0.5μl FcBlock (BD biosciences) and 0.5μl of markers and antibodies from BD bioscience. Samples were resuspended and incubated for 20 minutes in the dark and at room temperature. After that, wells were washed with FACS buffer (0.5% BSA in PBS), followed by fixation adding 150μl of 2% PFA.

Popliteal and footpad cells were harvested on days 7, 14, 21, and 28 post-infection. For surface staining analysis, a 200μl aliquot of the cell suspension described above was packed into a 96-well U-bottom culture dish. Next, cells were washed and stained for surface CD3, CD4, CD8, and CD44 using anti-CD3 BV786, anti-CD4 BUV395, anti-CD8 PE, and anti-CD44-PercP-Cy5.5 mAbs (Pharmingen).

For intracellular staining, cells were stimulated for 5 hours with anti-CD3 (clone 145-2C11; 0.1μg/ml) and anti-CD28 (clone 37.51; 1μg/ml) in the presence of Golgi-Stop (BD Biosciences). After cells had been washed and stained for surface CD4, CD25 and CD44 using anti-CD4 BUV395, anti-CD25 BV786, and anti-CD44-PercP-Cy5.5 mAbs (Pharmingen), they were fixed and permeabilized in Cytofix/ Cytoperm at 4˚C overnight. Permeabilized cells were stained with anti-IL-17A FITC, anti-IFN-γ Alexa 700 (clone XMG1.2), anti-IL-4 Alexa 647 and anti-Foxp3 PE-conjugated mAbs (Pharmingen) in FACS buffer for 30 min at 4˚C, washed, and analyzed by FACS. Cells were gated on live CD4+CD44$^{hi}$, and cytokine expression in each gate was analyzed (S1 Fig). The number of cytokine positive CD4+ T cells per footpad or popliteal was calculated by multiplying the percent of cytokine-producing cells by the number of CD4+ T cells found in footpad or popliteal.

## Neutralization/depletion assay with monoclonal antibodies

In order to evaluate the role of IL-17A, IFN-γ and CD25+ lymphocytes (mainly Treg) in the course of murine CBM, neutralization/depletion assays were performed by intravenous treatment every 3 days with 100μg of monoclonal antibodies anti-IL-17A (clone 17F3), anti-IFN-γ

(clone XMG1.2) and anti-CD25 (PC-61.5.3). As a control, we used IgG1 anti-mouse control (ICP) isotype (HRPN clone). Treatment with 20μg of antibodies was also performed intralesionally every 3 days. Anti-IL-17A (clone 17F3), Anti-IFN-γ (clone XMG1.2) and anti-CD25 (PC61.5.3) were purchased from BioXCell (West Lebanon, USA).

### Reporter cell assay and stimulation with *F. pedrosoi* morphotypes

B3Z T cells, bearing an NFAT-lacZ construct, were provided by Dr. Nilabh Shastri (University of California, Berkeley, CA). The construction and use of B3Z cells expressing Dectin-3 (also referred to as MCL, Clecsf8 or Clec4d), Dectin-2 or FcR, as well as BWZ cells expressing Dectin-1, have been described previously [27–31]. B3Z / BWZ cells were stimulated in a 96-well plate with *F. pedrosoi* conidia, fragments of hyphae and muriform cells at multiplicity of infection (MOI) of 1. Next, $10^5$ cells were incubated for 18 hours at 37˚ C and the activity of LacZ was measured after complete cell loading using Chlorophenol red-β-D-galactopyranoside-CPRG (Roche) as substrate. The plate was read in a microplate reader (OD 560 nM).

### Statistical analysis

Differences in the number or frequency of cytokine-producing T cells, as well as in the recovery of colony-forming units between experimental groups, were analyzed using the t-test for comparison between two populations, or analysis of variance (ANOVA) followed by the Bonferroni post-test method, performed in GraphPad Prism statistical software, version 6.0, GraphPad Software, San Diego, California, USA. Data were considered significant when p <0.05. All data used to draw the conclusions outlined in this work is available in S1 Dataset.

## Results

### Infection with *F. pedrosoi* propagules induces CD4+ T helper lymphocytes' migration into the site of infection (footpad)

T lymphocyte population analysis of a self-healing CBM mouse model did not show any changes in population balance between CD4+ and CD8 + T cells in dLN (popliteal), although, as expected, the total number of T lymphocytes increased over time (Fig 1A). In contrast, frequency and total number of CD4+ T cells increased significantly in the footpad following infection, compared to CD8 + T cells (Fig 1B and S2 Fig), indicating a possible role of those cells in the immune response during CBM.

### The evolution of T helper subsets in infected footpads and dLN during CBM

In order to evaluate subpopulations of CD4+ T lymphocytes in the course of CBM, total lymphocytes were isolated from dLN (popliteal) and infection site (footpad), followed by the analysis of T helper phenotypes.

The number of Th1 (CD4hi IFN-γ+) cells in the footpad increased steadily over 21 days post-infection and then decreased (Fig 1C and S3C Fig), whereas the expansion of Th1 cells was less pronounced in popliteal lymph node (Fig 1C and S4C Fig). The numbers of Th17 cells (CD4hi IL-17+) rose sharply during the early stages of the disease (until 14 days after infection) and fell gradually during later time points in both footpad (Fig 1D and S3B Fig) and dLN (Fig 1D and S4B Fig). Consequently, the Th1/Th17 ratio is lowest and highest in the footpad at days 7 and 21 post-infection, respectively (Fig 1E and 1F). Foxp3+ Treg cells also expand during the infection, being more numerous in dLN (Fig 1G). The relative frequencies in the footpad early in the infection are reduced compared to Th17 cells (S3B Fig). IL-4 producer cells

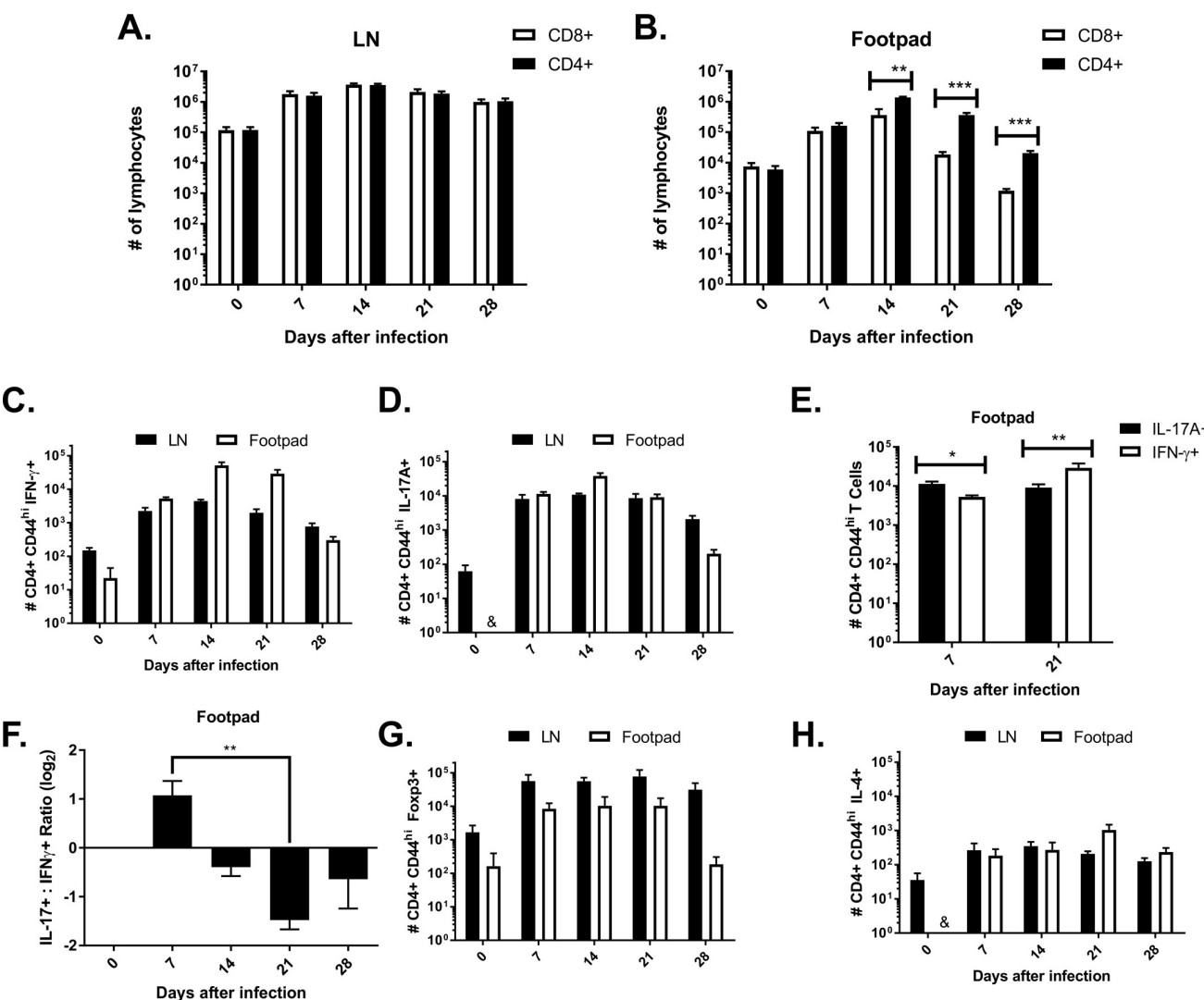

**Fig 1. T lymphocyte subset quantification in footpad and dLN of mice infected with *F. pedrosoi*.** CD4+ and CD8+ increase in popliteal lymph node (LN) after infection with $10^6$ *F. pedrosoi* propagules containing hyphae and conidia (**A**). Quantification of T lymphocytes in the footpad showed high numbers of CD4+ compared to CD8+ T cell (**B**). Significant expansion of IFN-γ+ (**C**), IL-17+ (**D**), and Foxp3+ (**G**) but not IL-4+ CD4+ T cells (**H**) was observed in dLN as well as in the site of infection during 28 days of infection with *F. pedrosoi*. In the footpad, the IL-17+ population was higher than IFN-γ + cells in the first 7 days of infection (**E**). The opposite occurs after 21 days, when the IFN-γ+ population is higher than IL-17+ T cells (**F**), reversing the ratio between these populations. * P <0.05, ** P <0.01.

failed to expand significantly in both footpad (Fig 1H and S3D Fig) and dLN (Fig 1H and S4D Fig).

Taking into consideration that this is a self-healing animal model of CBM, these results indicate that the increase in the Th17 population in the infection site during the first 14 days, followed by Th1 population increase, closely assisted by a growing Treg population until disease resolution, may represent the ideal balance of these populations when aiming at CBM remission.

## Depletion of CD25 + cells reduced CFUs in the footpad at day 14 post-infection but delayed disease resolution

To evaluate possible roles of Treg cells in the course of murine CBM, infected mice were injected with anti-CD25 monoclonal antibodies intravenously and into footpad lesions. To

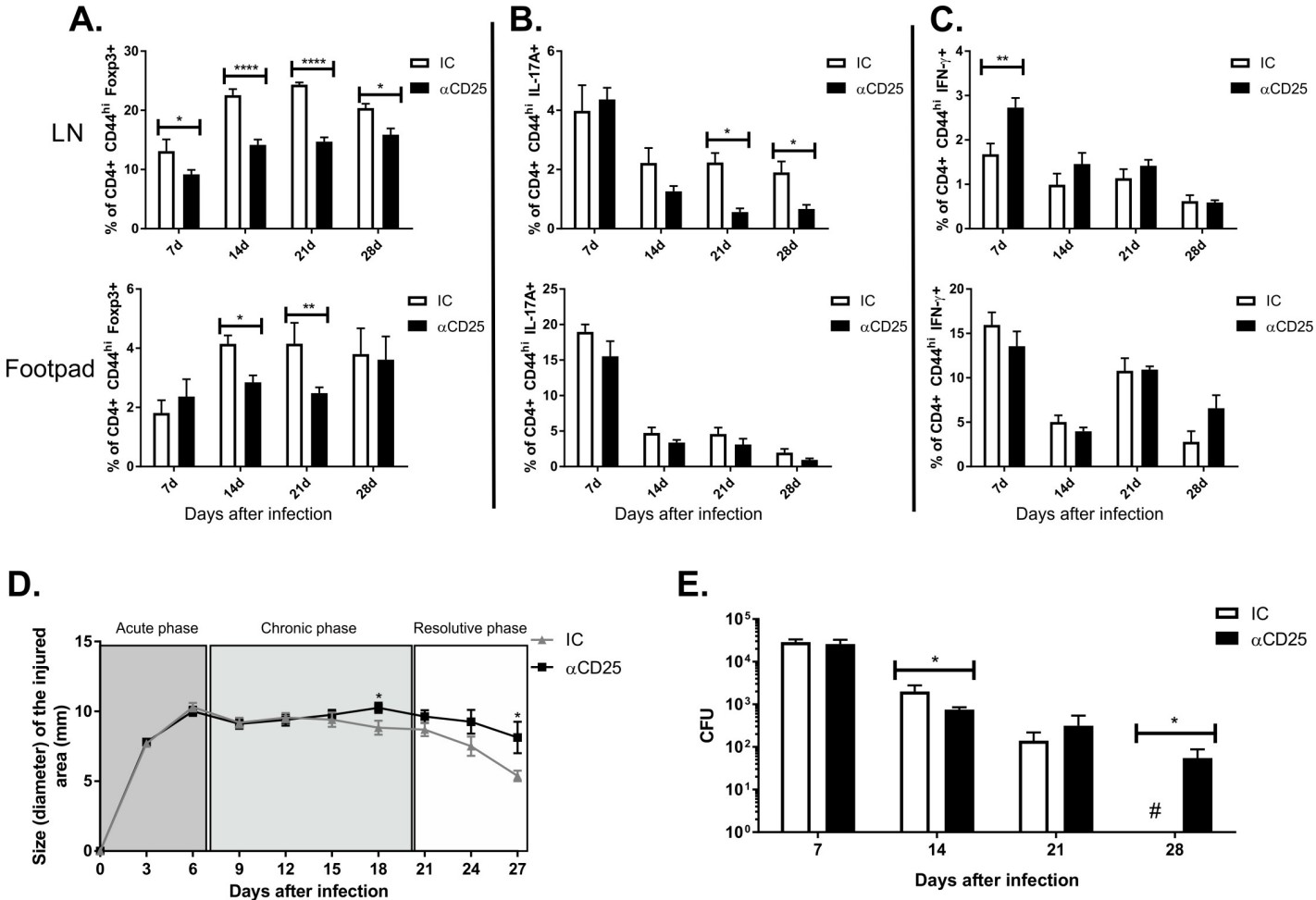

**Fig 2. Reduction of Treg population is related to fungus persistence in infected tissue and opposes self-healing in murine experimental CBM.** Low frequency of active CD4+ T lymphocytes expressing Foxp3 of animals treated with α-CD25 or control isotype (IC) is shown in popliteal lymph node and footpad (**A**). Frequency of IL-17+ T cells is decreased in dLN of animals treated with αCD25 after 21 and 28 days of infection (**B**). Frequency of IFN-γ+ T cells increased after 7 days of infection in animals treated with αCD25 when compared to control mice (**C**). Morphometric analysis showed increased inflammatory aspect in those animals treated with anti-CD25, as indicated by footpad swelling measures (**D**). CFU quantification displayed a small increase in fungal load in those animals after 14 days of infection; however, fungal clearance was impaired after 28 days (**E**).* P <0.05, ** P <0.01, *** P <0.001 and **** P<0.0001.

assess whether anti-CD25 mAbs depleted Treg cells, we enumerated CD4+ CD25+Foxp3+ cells from the dLN by FACS. The frequencies of Treg cells were significantly reduced in the course of infection, especially in dLN (Fig 2A and S5 Fig). However, anti-CD25 treatment did not significantly alter the frequency of Th1 and Th17 in the footpad during the infection (Fig 2B and 2C). Surprisingly, treatment with αCD25 slightly reduced frequencies of the Th17 population after 21 and 28 days of infection in dLN, while the Th1 population increased in the first 7 days of infection (Fig 2B and 2C).

The inflammatory response, as measured by footpad swelling, increased in mice treated with anti-CD25 mAb vs. mice treated with IgG control Ab (Fig 2D) after 18 days of infection. Footpad CFUs decreased significantly at day 14 post-infection (Fig 2E). However, mice treated with anti-CD25 mAb were not able to reduce their fungal load during the late stage of the infection, favoring the persistence of the fungus in the tissue (Fig 2E).

## Neutralization of IL-17A and IFN-γ impairs the elimination of fungal cells in the course of murine CBM

To investigate the relevance of Th17 and Th1 responses in the course of murine CBM, mice were infected with *F. pedrosoi* propagules and treated every three days with an anti-IL-17A monoclonal antibody and/or anti-IFN-γ by delivering the antibodies both intravenously and directly into lesions in the footpad. Neutralization of IL-17 reduced the inflammatory profile in the course of infection, as assessed by histopathological analysis (Fig 3A–3C) when compared to IC group treatment control (S6A Fig), whereas neutralization with anti-IFN-γ slightly intensified the inflammation at day 14, as assessed by histopathological analysis without being reflected in footpad swelling (Fig 3A, 3E and 3F). Combined treatment with both antibodies did not have an additive effect on inflammatory responses (Fig 3A, 3H and 3I).

Neutralization of IL-17 highly increased CFUs at day 7 post-infection (Fig 3D). In turn, neutralization of IFN-γ increased CFUs at days 7 and 14 (Fig 3G). Although not found to be statistically significant, neutralization of IFN-γ showed a trend towards increased CFUs vs. control mice at later time points, allowing fungal persistence up to 28 days after infection. Mice treated with anti-IFN-γ but not control mice exhibited detectable CFUs at the latest time point analyzed (S6B and S6C Fig). Combined treatment with both antibodies increased footpad CFUs at all time points analyzed (Fig 3J).

Similarly, IFN-γ KO animals failed to clear fungal infection by day 35 post-infection (S6D and S6E Fig).

These data suggest that Th17 cells might contribute to fungal elimination at the early stages of infection, whereas Th1 cells might be needed during the early and later stages of infection to eliminate the fungal pathogen.

## Dectin-2$^{-/-}$ mice have reduced Th17 cell responses and higher fungal burden at the chronic phase of experimental CBM

We previously reported that conidia of *F. pedrosoi* were able to induce Th17 cell differentiation in mice via Dectin-2 recognition [31,32]. To evaluate which CLRs recognize various *F. pedrosoi* morphotypes, we cultured CLR expressing reporter cells with spores, hyphae, and muriform cells. All three forms of *F. pedrosoi* stimulated Dectin-2 and to a lesser extent Dectin-1 signaling. Hyphae and muriform cell recognition by Dectin-2 were, in fact, more potent than conidia (S7 Fig).

Considering that dectin-2 is directly associated with Th17 lymphocyte differentiation, we hypothesized that dectin-2 KO animals would be more susceptible to *F. pedrosoi* infection, especially in the early stage of CBM infection. As expected, infected Dectin-2$^{-/-}$ mice showed a reduction in numbers and frequencies of Th17 cells in the footpad and dLN in the first 21 days of infection (Fig 4A). Inconsistent or no changes were found for the Treg and Th1 population in infected Dectin-2$^{-/-}$ vs. wild-type control mice (Fig 4B and 4C).

Morphometric analysis in Dectin-2$^{-/-}$ animals showed a significant reduction in the intensity of the inflammatory response as measured by footpad swelling throughout infection (Fig 4D). CFUs were reduced in Dectin-2$^{-/-}$ vs. wild-type mice at day 14 post-infection but were not impaired at the other time points (Fig 4E), highlighting the relevance of the Dectin-2 mediated Th17 population in controlling fungal burden in this stage of the disease.

## Discussion

Commonly, CBM develops after transcutaneous inoculation of fungal propagules such as hyphae fragments and conidia, usually as a result of traumas by contaminated parts of plants

**A.**

| Footpad Histopathological Assessment of Inflammation Levels | | | | |
|---|---|---|---|---|
| Treatment | 7 days | 14 days | 21 days | 28 days |
| (IC) | ++++ | +++ | ++ | + |
| αIL-17A | +++ | ++ | + | - |
| αIFN-γ | ++++ | ++++ | ++ | + |
| αIL-17A + αIFN-γ | +++ | ++ | ++ | + |

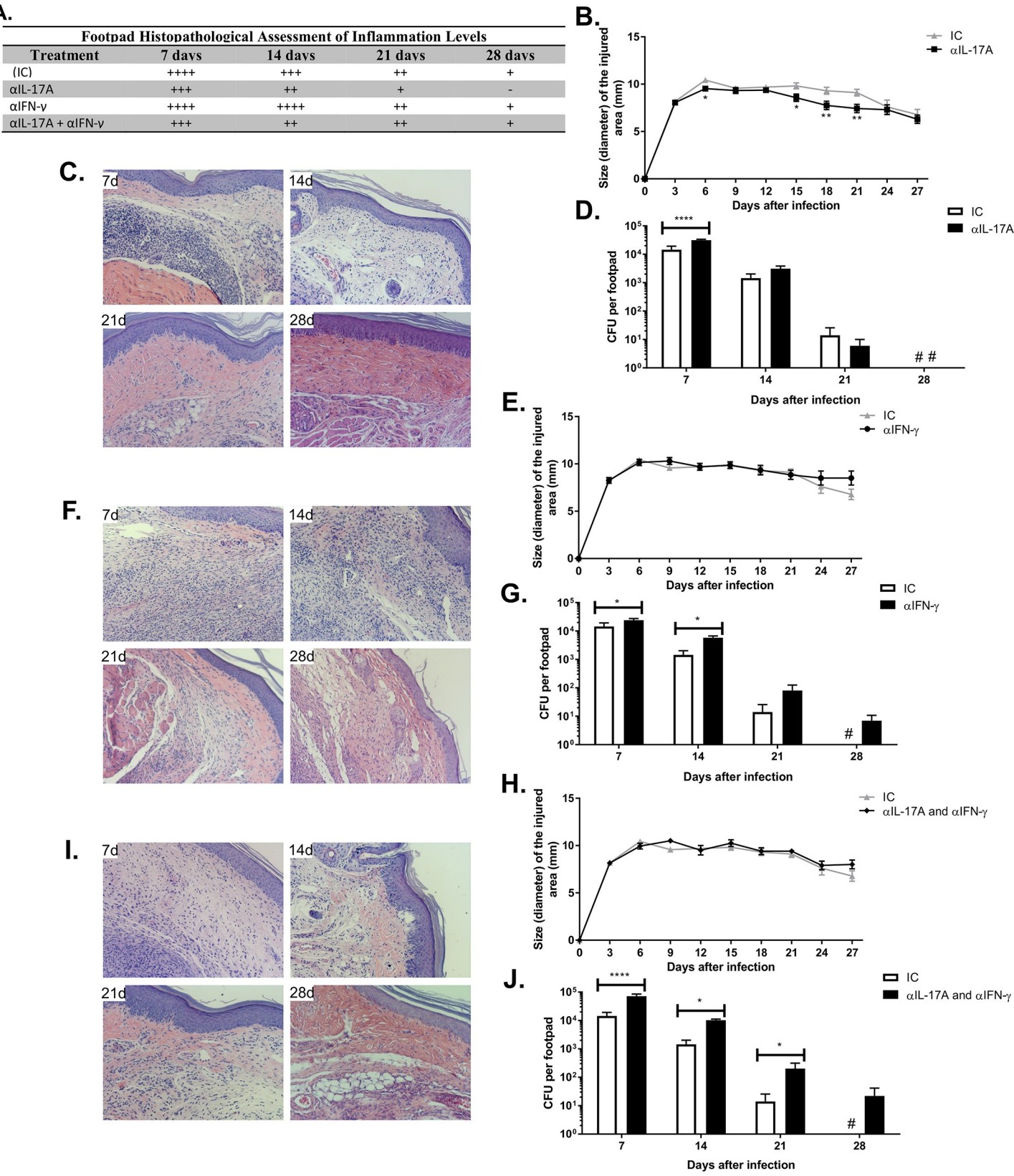

**Fig 3. Neutralization of IL-17A and IFN-γ leads to disturbance in the inflammatory response and increased fungal load.** Footpad histopathological assessment of inflammation (**A**) in animals treated with αIL-17A revealed a decrease in inflammation levels during CBM (**C**). For those animals treated with αIFN-γ, these inflammatory aspects increased in the first 14 days (**F**). Intermediate characteristics were detected in animals treated with both antibodies (**I**). Reduction in footpad swelling was observed in animals treated with αIL-17 after 15, 18, and 21 days of infection, when compared to an isotype control (IC), treated mice (**B**). Not much difference was observed in footpad swelling of animals treated with αIFN-γ (**E**) and with both antibodies (**H**). CFU quantification revealed an increase of fungal load in the first 7 days of infection in animals treated with αIL-17 (**D**). Increase in fungal loads was also observed in those mice treated with αIFN-γ alone (**G**) or in combination with αIL-17 (**J**), in which fungal clearance was impaired after 28 days of infection. HE staining, 200x magnification. * P <0.05, ** P <0.01, *** P <0.001 and **** P<0.0001.

such as thorns and wood chips [33]. Lower limbs are frequently affected and, once installed in the tissue, fungal propagules adhere to epithelial cells and differentiate into characteristic parasitic structures, called muriform cells [6,34,35].

Contrary to what occurs in patients who develop the disease, which maintains the fungal load and chronic inflammatory process for long periods of time, most murine models described to date tend to remission after a short period of infection (approximately after 30

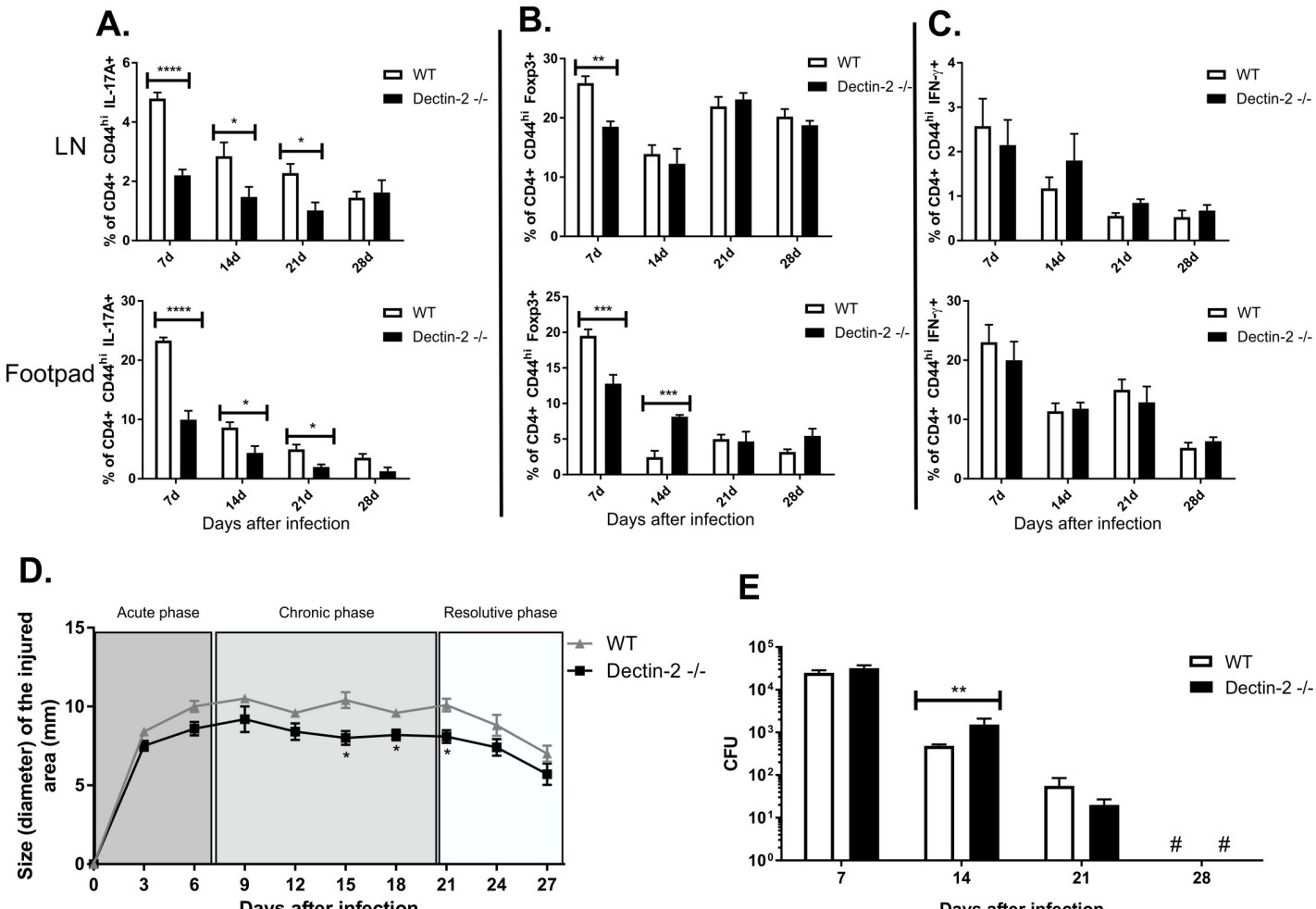

**Fig 4. IL-17+ cell expansion requires Dectin-2 and is relevant in fungal elimination in the early stages of experimental CBM.** Frequency of IL-17+ T cells is strongly decreased in dLN and footpad of Dectin-2 KO mice (**A**). The low frequency of activated CD4 + T lymphocytes expressing Foxp3 in dLN and footpad of Dectin-2 KO animals was observed in early stages of infection when compared to wild type mice (**B**). No significant changes were observed in the IFN-γ+ population in the course of infection in Dectin-2 KO animals when compared to wild type (**C**). Morphometric analysis showed a reduction in inflammatory aspect in Dectin-2 KO animals, as indicated by footpad swelling measures (**D**). CFU quantification displayed an increase in fungal load in those animals after 14 days of infection without impairing fungal clearance in later stages of infection (**E**).* P <0.05, ** P <0.01, *** P <0.001 and **** P<0.0001.

days of infection). What is observed in the murine model used in the present work is the establishment of an effective response pattern capable of dealing with the presence of different fungal forms, leading to disease remission without the occurrence of long-term chronicity as observed in humans.

To understand the balance between immune surveillance, disease progression, host invasion, and pathology, it is essential to be able to define the nature of the protective immune response to fungal invaders and other factors that predispose us to infection [3].

A state of chronic or intractable fungal disease may be the result of an unbalanced immune response which compromises the host's ability to cope with fungal infective cells rather than an "intrinsic" susceptibility to infection [36]. It has been affirmed that the absence of CD4+T cells impairs host defense against *F. pedrosoi* infection in mice [37]. In the infected footpad, the coordinated action of CD4+ T lymphocyte subpopulations was observed in this work, so that initial polarization of Th17 is followed in later stages by Th1 cells, accompanied by a high frequency of CD4 + Foxp3 + (Fig 1C–1G). Note that Th17 polarization in the early stage of infection does not mean a lack of Th1 response; on the contrary, the data presented demonstrate the relevance of the Th1 population already in the initial stages of infection (Fig 3G and S6D and S6E Fig). Many CD4+ T lymphocytes expressing Foxp3 were also observed in draining lymph node, favoring a regulated environment of the immune response (Fig 1G).

Th polarization seems to follow the transition of fungal forms in the course of experimental CBM in an attempt to deal correctly with the infection. We have already described the variation in fungal forms' frequency in infected tissue during the course of experimental murine CBM, with a higher number of hyphae and conidia in initial post-infection phase and many muriform cells in later stages of infection [38]. The same pattern was observed by Dong and co-workers in mice intraperitoneally infected with *F. pedrosoi* [39].

A reduction in the number of hyphae and conidia during the infection can be related to their transformation into muriform cells, as well as to their removal from tissue due to host effector mechanisms. Initial Th17 polarization was related to fungal load reduction at early stages of infection, as seen in the IL-17A neutralization assay (Fig 3D), in which fungal cells are composed mostly of fragments of hyphae and conidia. After 14 days of infection, when most fungal cells transformed into muriform cells, Th17 polarization was supplanted by the Th1 population, considered the most effective response pattern against muriform cells, which are usually localized within giant multinucleated cells.

Neutrophils represent the host's first line of defense in CBM, followed by activated macrophages, sometimes in the form of epithelioid cells and/or giant multinucleated cells [40]. IL-17 secreted by Th17 cells promotes neutrophils' maturation and migration to the site of infection, conferring protection against extracellular pathogens [41,42], whereas IFN-γ produced by the Th1 population properly activates neutrophils and enhances fungal cells' phagocytosis [43].

Since the elimination of conidia and muriform cells depends on phagocytosis, followed by proper activation of phagocytic cell effector mechanisms, it is not surprising that IFN-γ level reduction affects both early and later immune response against *F. pedrosoi*. Gimenes and co-workers demonstrated that patients with a severe form of CBM produce high levels of IL-10 and low levels of IFN-γ, together with inefficient T-cell proliferation. Meanwhile, patients with the mild form of the disease show intense production of IFN-γ, low levels of IL-10, and efficient T-cell proliferation [6,18]. Besides, it was already demonstrated that intraperitoneal administration of exogeneous IFN-γ significantly reduces the fungal load in the spleens of BALB/c mice infected with *F. pedrosoi* muriform cells and inhibits the peritoneal dissemination of this agent [39].

High doses of IL-17 in lesions essentially represent a host strategy against fungal infections. However, an average Th17 response may overlap the regulatory role of Tregs, and this

imbalance may eventually result in a less effective response to the fungus [16]. Thus, by controlling the quality and magnitude of the effector responses of innate and adaptive immunity, Treg cells may be responsible for a broad spectrum of host response, ranging from protective tolerance to notorious immunosuppression [43].

The population balance of Th must be adequate at each stage of the infection process, acting dynamically. Disturbances in this balance, as seen in the dectin-2 KO animal assay, show a reduced Th17 population and impaired fungal elimination in the early stage of the disease without, however, compromising the disease remission process observed in the murine model after 28 days (Fig 4). On the other hand, disturbances involving IFN-γ neutralization in infected mice have hindered fungal elimination in both early and later stages of infection (Fig 3G).

Immune responses mediated by CLRs include phagocytosis, induction of antifungal effector mechanisms, and the production of various soluble mediators, including cytokines, chemokines, and inflammatory lipids [44]. In addition, these receptors are responsible for directing and modulating the development of the adaptive immune response, especially Th1 and Th17 [45,46]. Not only are *F. pedrosoi* conidia recognized by Dectin-2, as we previously reported [31] but also fungal hyphae, and muriform cells can bind this receptor, promoting Th17 expansion (S7 Fig, Fig 4A). We also reported that *F. pedrosoi* recognition by dectin-1 and dectin-2 increases populations of IL-17 secreting cells [31].

Th17 cells may act as protagonists in the immunity against infectious agents in inflammatory conditions, since this population, at least in part, can enhance the immune response, acting in concomitance with Th1 and Th2 response patterns [47,48]. However, recent studies have shown that IL-17-producing lymphocytes in association with IL-23 production are involved in autoimmune damage caused by experimental allergic encephalomyelitis, collagen-induced arthritis, and inflammatory bowel disease [49]. IL-23 directs the development of Th17, promoting the development of chronic inflammatory processes dominated by the presence of IL-17, IL-6, IL-8, and TNFα, as well as the intense activity of neutrophils and monocytes.

*In situ* studies with lesions of CBM patients revealed a higher quantification of IL-17 when compared to other mycoses such as paracoccidioidomycosis [16]. Not only that, but CBM lesions are also characterized by intense monocytic and neutrophilic response, as well as the presence of pro-inflammatory cytokines such as IL-1β, TNFα, and IL-6 [4,50,51].

Although the inflammatory response comprises a critical component in fungal immunity, its deregulation may be even worse in fungal infections. Both inflammation and infection itself are exacerbated when Th17 is upregulated in response to *C. albicans* and *A. fumigatus*. In these infection models, IL-23 and IL-17 subverted neutrophil-mediated immunity, resulting in severe inflammatory tissue pathology associated with infection [48,52–56]. A recent study by Silva and co-workers identified high expression of IL-17 secreting T-cells on cell infiltrates of CBM human lesions, concomitantly with an insignificant presence of lymphocytes displaying a regulatory phenotype [16].

Regulatory T cells (Treg), characterized by high expression of CD25 and controlled by Foxp3 transcription factor, are responsible for limiting autoimmunity and chronic inflammation [57]. Therefore, investigating the performance of this population in the scope of CBM becomes exceptionally important, since the disease in humans goes through a chronic inflammatory process, becoming unable to eliminate fungal cells from skin lesions [58]

The anti-inflammatory activity of Tregs has been described in fungal infections in murine models and humans. In experimental models of fungal infections, both inflammation and tolerance are controlled by the coordinated action of Tregs [43]. However, considering that Treg-mediated response may limit the effectiveness of the protective immune response, when not

followed by an active cellular response such as Th1, the consequence may be the persistence of fungal infection [59,60]. Failure to control the intense inflammatory response, induced for example by the presence of muriform cells in the late stage of infection [38], may be related to failure to modulate host immunological response, allowing fungal permanence in the tissue and disease chronification. It should be considered, then, that any failure in Tregs, favoring upregulated Th17 activity, coupled with an inefficient Th1 response, may be determining factors for the development of CBM in humans.

In summary, our work expands long-standing Th1/Th2 dualized comprehension of CBM immunopathogenicity, showing the relevance of Dectin-2 mediated Th17 and Treg cells in a self-healing model of CBM. In this context, it has been demonstrated that both Th1 and Th17 responses, with the aid of Treg cells, are required for appropriate immune response in CBM, varying in intensity throughout the immune response in the course of the disease.

By uncovering the basic cellular mechanisms that are responsible for the development of immunopathology and host control of fungal disease, this information will provide a foundation for new treatment strategies, not only for CBM but also for other mycoses.

## Supporting information

**S1 Dataset. Raw data used to draw the conclusions outlined in this work.**
(XLSX)

**S1 Fig.** In order to select a relevant T lymphocyte population for proper investigation, a gate strategy was built, aiming to select only lymphocytes, according to their morphometric aspects (**A**), which were isolated (**B**), live (**C**) and activated (**D**). Live CD4+ was established using Live and Dead dye, and activated CD4+ cells were then gated considering a hi expression of CD44.
(TIF)

**S2 Fig.** Cytometry dotplots aimed to identify and quantify CD8+ and CD4+ T cells in the footpad and draining lymph node (LN) in the course of experimental CBM (**B**). Uninfected animals were used as control (**A**).
(TIF)

**S3 Fig.** Cytometry dotplots aimed to identify Foxp3+, IL-17A+, IFN-Y+, and IL-4+ CD4+ T cells' subpopulation in the footpad of animals infected with *F. pedrosoi* in the course of experimental CBM (**B**). Uninfected animals were used as control (**A**).
(TIF)

**S4 Fig.** Cytometry dotplots aimed to identify Foxp3+, IL-17A+, IFN-Y+, and IL-4+ CD4+ T cells' subpopulation in draining lymph node (LN) in the course of experimental CBM (**B**). Uninfected animals were used as control (**A**).
(TIF)

**S5 Fig. Density plots in order to quantify the Treg population in animals treated with αCD25 when compared to an isotype control (IC).**
(TIF)

**S6 Fig.** Histopathology of animals treated with isotype control and used as a control group for inflammation level measures, HE staining and 200x magnification (**A**). Histopathology of animals treated with αIFN-γ after 28 days of infection is displayed, showing the presence of muriform cells (arrows) in 200x (**B**) and 400x magnification (**C**). CFU quantification in IFN-γ -/- animals shows impaired fungal clearance after 28 and 35 days of infection (**D-E**).
(TIF)

**S7 Fig. *F. pedrosoi* fungal forms are recognized by dectin-2 and dectin-1.** Interaction test between fungal forms with reporter cells expressing dectin-1 (**B**), dectin-2 (**D**), dectin-3 (**E**) and mincle (**F**) and carrying NFAT-lacZ construct was evaluated. Cells not expressing CRL (**A**) or expressing only FcR (**C**) were used as controls. * P <0.05 and *** P <0.001. (TIF)

## Acknowledgments

The authors would like to thank Viviane Montanaro Leal, for technical assistance in the histochemical assay. The authors also thank University of Wisconsin Carbone Cancer Center (WCCC) Flow Lab facility and personnel, for technical support.

## Author Contributions

**Conceptualization:** Bruce Klein, Anamelia Lorenzetti Bocca.

**Data curation:** Marcel Wüthrich.

**Formal analysis:** Isaque Medeiros Siqueira, Anamelia Lorenzetti Bocca.

**Funding acquisition:** Bruce Klein.

**Investigation:** Isaque Medeiros Siqueira, Mengyi Li, Huafeng Wang, Raffael Júnio Araújo de Castro.

**Methodology:** Isaque Medeiros Siqueira, Lucas de Oliveira Las-Casas, Raffael Júnio Araújo de Castro.

**Project administration:** Anamelia Lorenzetti Bocca.

**Supervision:** Marcel Wüthrich, Bruce Klein, Anamelia Lorenzetti Bocca.

**Writing – original draft:** Isaque Medeiros Siqueira.

**Writing – review & editing:** Marcel Wüthrich, Anamelia Lorenzetti Bocca.

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
