## [Decision Letter · Decision Letter 0]

3 Mar 2020

Dear Dr Bocca,

Thank you very much for submitting your manuscript "Early immune response against Fonseca Pedroso requires Dectin-2 mediated Th17 activity, whereas Th1 response, aided Treg cells, is crucial for fungal clearance in later stage of experimental chromoblastomycosis." for consideration at PLOS Neglected Tropical Diseases. As with all papers reviewed by the journal, your manuscript was reviewed by members of the editorial board and by several independent reviewers. The reviewers appreciated the attention to an important topic. Based on the reviews, we are likely to accept this manuscript for publication, providing that you modify the manuscript according to the review recommendations. 

Sincerely,

Todd B. Reynolds

Deputy Editor

Todd Reynolds

Deputy Editor

Reviewer's Responses to Questions

**Key Review Criteria Required for Acceptance?**

**Methods**

-Are the objectives of the study clearly articulated with a clear testable hypothesis stated?

-Is the study design appropriate to address the stated objectives?

-Is the population clearly described and appropriate for the hypothesis being tested?

-Is the sample size sufficient to ensure adequate power to address the hypothesis being tested?

-Were correct statistical analysis used to support conclusions?

-Are there concerns about ethical or regulatory requirements being met?

Reviewer #1: The objectives of the study are clearly articulated with an appropriate design . The authors demonstrate to have the sample size sufficient to answer the hypothesis tested with statistical analysis to support the conclusions. The unique question is about ethical requirements concerning to the animal ethical committee permission. In case it was done, I suggest to present it in the methodology session.

Reviewer #2: The methods are clear and well done.

**Results**

-Does the analysis presented match the analysis plan?

-Are the results clearly and completely presented?

-Are the figures (Tables, Images) of sufficient quality for clarity?

Reviewer #1: The results is clear and respond the main questions proposed.

Reviewer #2: Table and figures are clear.

Results are well exposed.

**Conclusions**

-Are the conclusions supported by the data presented?

-Are the limitations of analysis clearly described?

-Do the authors discuss how these data can be helpful to advance our understanding of the topic under study?

-Is public health relevance addressed?

Reviewer #1: The manuscript is in general well written and contains important information regarding the early immune response against Fonsecaea pedrosoi demonstrating by the animal assays, showing that Th1 response, aided Treg cells, is crucial for fungal clearance in later stage of experimental chromoblastomycosis. The unique concern is why the authors didn't test the related species that are also considered causal agents of the disease? Such as, e.g., F. monophora and or Cladophialophora carrionii? Even though other species were not included in the study, I suggest to have at least one a paragraph in the discussion session about the disease clinic variations, if is it exclusively caused by the host's responses or not? A paragraph about this issue would be interesting, specially, as future perspective of this work in order to test other species and clarify more the immunology profile of this disease. The other issue is that the authors inoculated the animals with muriforms cells produced in vitro. Do the authors have previous experience or tested on differences about response using fungal propagules for inoculation instead of muriform cells? If yes could you please discuss it?

Reviewer #2: Conclusions are fully supported by the data

**Editorial and Data Presentation Modifications?**

Reviewer #1: I recommend a minor revision as following below: 

1- Authors should review the abbreviation of the volume unit used and standardize it. For example, throughout the text the authors used L and l for liter ( see e.g. the line 160 that was used mL and μl). I suggest using L for liters (e.g. μL and mL) and "l"only for chemical formulas;

3- In the methodology is necessary to provide the register number of the approval from the animals ethics committee. 

4- The authors must do a review of the scientific names, which must be in italics in all sections of the manuscript. In the reference section there are several scientific names that are not in Italic.

5- I suggest to have at least one paragraph in the discussion session about the disease clinic variations, if is it exclusively caused by the host's responses or not? A paragraph about this issue would be interesting, specially, as future perspective of this work, in order to test other species and clarify more about the immunology profile of this disease.

Reviewer #2: Discussion needs to be broadened based on recent literature advances on the subject

**Summary and General Comments**

Reviewer #1: The work is showing the relevance of Dectin-2 mediated Th17 and Treg cells in a self-healing model of CBM. It has been demonstrated that Th17 responses with the aid of Treg cells are required for appropriate immune response in CBM, varying in intensity throughout the immune response in the course of the disease associated to the F. pedrosoi. In general it is a relevant article and it is well written with only minor suggested corrections and some questions concerning to the strains tested and inoculum form that could be stressed on the discussion session.

Reviewer #2: This paper is well done and provides interesting insight in the immunity against F pedrosoi in mice who are able to cure spontaneously the infection as opposed to a chronic infection in humans.

More discussion about the relation between the author's findings on TH1 and TH17 and the reference 

Sousa, M., Reid, D., Schweighoffer, E., Tybulewicz, V., Ruland, J., Langhorne, J., Yamasaki, S., Taylor, P., Almeida, S., Brown, G. (2011). Restoration of Pattern Recognition Receptor Costimulation to Treat Chromoblastomycosis, a Chronic Fungal Infection of the Skin Cell Host & Microbe 9(5), 436-443. https://dx.doi.org/10.1016/j.chom.2011.04.005

and

Sousa, M., Belda, W., Spina, R., Lota, P., Valente, N., Brown, G., Criado, P., Benard, G. (2014). Topical Application of Imiquimod as a Treatment for Chromoblastomycosis Clinical Infectious Diseases 58(12), 1734-1737. https://dx.doi.org/10.1093/cid/ciu168

where the activation of some PRR via TLR ligands or imiquimod was able to help curing the disease.

PLOS authors have the option to publish the peer review history of their article (what does this mean?). If published, this will include your full peer review and any attached files.

Reviewer #1: No

Reviewer #2: No
---

## [Editor Report · Decision Letter 1]

12 May 2020

Dear Dr Bocca,

We are pleased to inform you that your manuscript 'Early immune response against Fonseca Pedroso requires Dectin-2 mediated Th17 activity, whereas Th1 response, aided Treg cells, is crucial for fungal clearance in later stage of experimental chromoblastomycosis.' has been provisionally accepted for publication in PLOS Neglected Tropical Diseases.

Best regards,

Todd B. Reynolds

Deputy Editor

Todd Reynolds

Deputy Editor

---

## [Editor Report · Acceptance letter]

8 Jun 2020

Dear Dr Bocca,

We are delighted to inform you that your manuscript, "Early immune response against *Fonsecaea pedrosoi* requires Dectin-2-mediated Th17 activity, whereas Th1 response, aided by Treg cells, is crucial for fungal clearance in later stage of experimental chromoblastomycosis," has been formally accepted for publication in PLOS Neglected Tropical Diseases.

Best regards,

Serap Aksoy

Editor-in-Chief

Shaden Kamhawi

Editor-in-Chief
